



# Calculating the sediment budget of a tropical lake in the Blue Nile basin: Lake Tana

F. A. Zimale[1], M. A. Moges[1], M. L. Alemu[1], E. K. Ayana[2,3], S. S. Demissie[4], S. A. Tilahun[2], T. S. Steenhuis[2,5*]

[1] School of Civil and Water Resources Engineering, PhD program in Integrated Water Management, Bahir Dar Institute of Technology, Bahir Dar University, Bahir Dar, Ethiopia

[2] School of Civil and Water Resources Engineering, Bahir Dar Institute of Technology, Bahir Dar University, Bahir Dar, Ethiopia

[3]Department of Ecology, Evolution and Environmental Biology, Columbia University, New York, NY, USA

[4]Ethiopian Institute of Water Resources, Addis Ababa, Ethiopia

[5]Department of Biological and Environmental Engineering, Cornell University, Ithaca, NY, USA

* *Correspondence to*: Tammo S. Steenhuis (tss1@cornell.edu)

**Abstract:** Soil erosion decreases soil fertility of the uplands and causes siltation of lakes and reservoirs. However, very little data exists to quantify accurately the impact of sediment on lakes in tropical monsoonal areas in the African highlands. Lake Tana is one of these lakes in Ethiopia. The objective of this study is to quantify the sediment budget for Lake Tana watershed with limited observational data. To overcome these limitations we use the Parameter Efficient Distributed (PED) model that has shown to perform well in the Ethiopian highlands. PED model parameters are calibrated using daily discharge data and sediment concentration infrequently measured for establishing sediment rating curves for the major rivers. The calibrated model parameters are then used to predict the sediment budget for the period 1994-2009. Sediment retained in the lake is calculated from two bathymetric taken 15 years apart and the sediment leaving the lake is based on measured discharge and observed sediment concentrations. Results show that annually on average 34 Mg/ha/year of sediment is removed from the gauged part of the Lake Tana watersheds. Depending on the up scaling method, 14 to 32 Mg/ha/year is transported from the watershed of which 82% to 96% (with the upper estimate more likely) is trapped on the floodplains and in the lake.

Key words: Erosion, saturation excess, hydrology, discharge, floodplain, East Africa, Horn of Africa

## 1 Introduction

The Blue Nile basin drains approximately 16 % of Ethiopia and provides over half of the consumptive water use of Sudan and Egypt. The basin includes steep mountains and Lake Tana, the largest freshwater lake in Ethiopia is home to a unique fish population of which twenty fish species are endemic (Vijverberg *et al.*, 2009). In the 1920's Major R.E Cheesman, a cartographer and British representative to Northwest Ethiopia based in Dangila described the lake as beautiful, pristine with a sandy bottom near the inlets of the major rivers (Cheesman, 1936). Currently, the lake water has become polluted with sediment and nutrients. The sandy bottoms near the inlets have been replaced by newly formed deltas which in the case of the Gilgel Abay River is 10 km long. The decline in the lake water quality affects the livelihood of over 500,000 people that



are directly or indirectly dependent on the lake and its wetlands (Vijverberg *et al.*, 2009). Most notably increased turbidity due to sediment reduces water light intensity, threatening the aquatic ecosystem. Understanding the sediment dynamics and particularly the amount of sediment reaching Lake Tana is crucial for better management of the lake and its watershed.

In semi-arid northern Ethiopia, sediment dynamics and gully formation have been well documented (Gebrermichael *et al.*,
2005; Aerts *et al.*, 2006; Descheemaeker *et al.*, 2006; Gebreegziabher *et al.*, 2009; Girmay *et al.*, 2009; Frankl *et al.*, 2011; Frankl *et al.*, 2013; Haregeweyn *et al.*, 2013). In the humid highlands, sediment dynamics have been less well studied and the information available mainly consists of data gathered in the Soil Conservation Research Program (SCRP) watersheds. For these relatively small watersheds (Guzman *et al.*, 2013) reported soil loss rate of 5.2 t ha$^{-1}$ yr$^{-1}$ (Andit Tid), 24.7 t ha$^{-1}$ yr$^{-1}$ (Anjeni ) and 7.4 t ha$^{-1}$ yr$^{-1}$ (Maybar). Greater soil loss ranges are reported on test plots with 32 to 36 t ha$^{-1}$ yr$^{-1}$ for
Maybar, 87 to 212 t ha$^{-1}$ yr$^{-1}$ for Andit Tid and 131 to 170 t ha$^{-1}$ yr$^{-1}$ for Anjeni watershed (Haile *et al.*, 2006a). Sediment loss as high as 540 t ha$^{-1}$ yr$^{-1}$ has been reported mainly due to gully erosion (Tebebu *et al.*, 2010). Sediment data from the sampling station on the Blue Nile at the border with Sudan indicated that current losses in the 180,000 km$^2$ basin are in the order of 7 ton/ha/yr (Yasir *et al.*, 2014). Thus most of the sediment that erodes from the land is deposited on its way to Sudan. Lake Tana is one of these sinks.

Lake management requires accurate discharge and sediment predictions. SWAT is often employed to simulate the discharge in the Ethiopian highlands (Setegn, 2008; Setegn *et al.*, 2009; Easton *et al.*, 2010; Setegn, 2010; Setegn *et al.*, 2010; Betrie *et al.*, 2011; Setegn *et al.*, 2011; Yasir *et al.*, 2014). Easton et al (2010) and White et al (2011) modified SWAT to include saturation excess runoff which is one of the major runoff mechanisms in the highlands. Water balance approaches have been utilized as well with some success (Conway, 2000; Mishra et al., 2004; Kebede et al., 2006; Steenhuis et al. 2009; Rientjes et
al., 2011; Tilahun et al 2015).

Early erosion predictions (Haregeweyn and Yohannes, 2003; Tamene *et al.*, 2006; Haile *et al.*, 2006b) in Ethiopia were either based on the sediment rating curve or the Universal Soil Loss Equation (USLE). More recently, sediment flows have been simulated in the Blue Nile basin employing various models: SWAT in which sediment predictions are based on MUSLE (Easton *et al.*, 2010; Setegn *et al.*, 2010; Betrie *et al.*, 2011; Yasir *et al.*, 2014), the modified SWAT-WB Water
Balance model (Easton et al., 2010), Parameter efficient model (PED) that is based on the Hairsine and Rose model (Hairsine and Rose, 1992; Steenhuis *et al.*, 2009; Tilahun et al 2013a, 2013b, 2015), WATEM/SEDEM (Haregeweyn et al. 2013) and Water Erosion Prediction Project (WEPP) model (Zeleke, 2000).

Only the studies of Easton *et al.* (2010) and Setegn *et al.* (2010) simulated sediment loads at the gaging stations near Lake Tana. Setegn et al. (2010) showed that sediment loads of 30 to 60 t ha$^{-1}$yr$^{-1}$ are exported from the Lake Tana watersheds
whereas Easton et al. (2010) predicted that a maximum of 84 t ha$^{-1}$yr$^{-1}$ can be exported from the Gumara watershed. Kaba *et al.* (2014) determined the sediment contribution from Gumara watershed for a 10 year period using MODIS satellite

Imagery. These lake concentrations were an order of magnitude less than those observed concentrations in the rivers at the gauges. The only estimate on sediment accumulated or being transported into the lake available to date  is by Ayana (2007) who analyzed two bathymetric surveys (1987 and 2006) of Lake Tana  and found that 200 Mm$^3$ of sediment had settled at the bottom of the Lake during this 20 year period.

The objective of this study is combine current knowledge on sediment transport and to quantify the sediment budget for the Lake Tana and its watershed. Lake Tana is the largest lake in Ethiopia.

## 2 Material and methods

### 2.1 Study area:  the Lake Tana basin

Lake Tana is the largest lake in Ethiopia. It is in the headwaters of the Blue Nile River and has a drainage area of

approximately 15,000 km$^2$.  The lake covers about 3,000 km$^2$ at an elevation of 1,786 m. The Lake is fed by four major gauged rivers: Gilgel Abay in the south, Megech in the North, Gumara and Rib in the east (Fig. 1). The Blue Nile exits the lake at the south end where the Chara Chara Weir has been constructed in 1995 to regulate the flow for hydropower generation. A brief summary of the major physiographic features of the contributing rivers and their watersheds is provided below.

Analysis of a 90m resolution digital elevation model (DEM) shows the slope in the Lake Tana Basin ranges from 0% to 39%. Altitude of the basin varies from 1785m to 4094m with a mean elevation of 2418m. Rainfall varies both in time and space and decreases to the north. The average dry season (November - April) rainfall is 117mm and potential evaporation is 710mm. The wet season (May-October) rainfall is 1400mm and potential evaporation is 645mm for data used in the calibration and validation periods. More than 90% of the annual rainfall occurs in the wet monsoon phase. The mean annual

temperature in the basin is 23 ºC in the relatively lower lying areas such as Bahir Dar and ranges between 15 to 20 ºC in the middle and high altitudes.

### 2.1.1 Gilgel Abay

The Gilgel Abay watershed is 4027 km$^2$. It constitutes 34% of the total sub basin  and contributes almost 60 % of the inflow to the Lake (Uhlenbrook *et al.*, 2010) .  Approximately 2767 km$^2$ is gauged (Gilgel Abay - 1665 km$^2$, Bered – 81 km$^2$, Koga

25   - 244 km$^2$ and Kilti – 767 km$^2$). Almost 50% of the hillslopes have an inclination between 2% - 8% and represents about 50% of the total area (Fig. S1a in the supplementary material). The soil types are Haplic Luvisols (56%) and Haplic Alisols (41%) which are volcanic from origin and have silty loam texture (Fig. S2 in the supplementary material). Approximately 74% is crop land (Fig. S3).



### 2.1.2 Gumara

Gumara watershed is found in the eastern part of the Lake Tana sub basin. The watershed at the lake inlet is 1970 km² and 65% of it is gauged. It originates at the foot of Mount Guna which is 4120m high. Sixty nine percent of the watershed has slopes of more than 8% (Fig. S1b in the supplementary material) and 96% is crop land (Fig. S3 in the supplementary material). The soils are mainly Haplic Luvisols (64%), Chromic Luvisols (24%), Eutric Leptosols (9%) and Eutric Vertisols (3%) (Fig. S2).

### 2.1.3 Rib

The Rib originates on Mount Guna and covers an area of 1668 km² at the lake inlet, of which 1289 km² is gauged. Thirty-seven percent of the gauged watershed has slopes of 0-8% and 63% has slopes of more than 8% (Fig. S1c). Compared to Gumara the Rib has more area with slopes greater than 30% (compare with Figs S1b and S1c). The slope at the lower portion of the watershed is nearly flat and the river bed shifts considerably. SMEC (2008) reported a more than 1.5m rise of the zero flow level at the gaging station in the years 1990 – 2002. In addition, the river bed is not stable at the gauging station (SMEC, 2008). As a result the carrying capacity of the river is reduced. The river overflows its banks regularly during the rainy monsoon phase and stream measurements are not conducted when the riverbank floods. Recent LANDSAT 8 images also showed a change in flow path by the river at the eastern flank of the lake. In the dry monsoon phase the river bed is dug out for obtaining sand. The holes fill up during the rainy phase. The dominant soils are Chromic Luvisols (40%), Eutric Leptosols (36%) and Eutric Fluvisols (24%) refer (Fig. S2). Ninety-one percent of the land is crop land and 9 % is grazing land (Fig. S3).

### 2.1.4 Megech

The Megech is located on the northern part of the Lake Tana sub-basin. It originates near the Simien Mountains at an altitude of around 4000 m. The total watershed is 663 km² at the lake inlet of which 500 km² is gauged. In 1997 a dam was constructed on a tributary of Megech River that supplies the town of Gondar with water. The reservoir has a surface area of 51 ha and a design capacity of 5.3 Mm³ with a catchment area of 68 km² (i.e., 13% of the gauged Megech watershed). Eighty-two percent of the catchment has slopes of more than 8% (Fig. S1d). Ninety-five percent of the catchment is crop land (Fig. S3) and Eutric Leptosols cover about 82% of the area (Fig. S2). In 2007 one-third of the volume of the reservoir was filled with sediment.

### 2.2 Available hydrological data

### 2.2.1 Meteorological data

The climate in the Lake Tana basin is affected by the movement of the inter-tropical convergence zone, which results in a single rainy season between June and September and a dry period in the rest of the year. Rainfall and temperature data were obtained from Ethiopian National Meteorological Agency (ENMA). Thiessen polygon method was applied on data obtained

from the surrounding meteorological stations to determine the areal rainfall data for each watershed. The potential evapotranspiration was calculated from minimum and maximum air temperature and other climate variables using the FAO Penman method with data from Bahir Dar weather station (Allen et al. 1998). Missing rainfall and temperature data were estimated using mean values of same dates in years with data.

### 2.2.2 Discharge

Daily stream discharge of Gilgel Abay, Gumara, Rib and Megech rivers were obtained from the Ministry of Water, Irrigation and Electricity of Ethiopia. Flow records from 1994 to 2009 were used for calibration and validation. Missing data were replaced with the mean of the available discharge data for the specific day.

### 2.2.3 Sediment concentrations

Ministry of Water, Irrigation and Electricity (MWIE) estimates sediment yield for the major rivers in the Lake Tana Basin using rating curves. These rating curves require periodic measurements of sediment concentration and discharge. Elevated sediment concentrations occur during the rainy phase in June, July and August. Daily sediment concentrations were determined by dividing the daily sediment load by the discharge measured during the same day.

MWIE sediment load data were available between 1964 and 2009. The distribution is uneven with some months without any data mainly during the dry monsoon phase when sediment concentrations are low. The Gumara and the Rib were monitored most intensively with just over 60 data pairs of sediment load and discharge. Sediment concentration of all rivers with the exception of the Gilgel Abay peaks before the discharge. This general trend of sediment concentrations is similar to the small experimental watersheds, Andit Tid, Maybar and Anjeni (Guzman *et al.*, 2013). According to Guzman *et al.* (2013) and Zegeye *et al.* (2010) the greatest concentrations occur at the time when the rills are formed in the newly ploughed agricultural lands. The sediment outflow from the Lake peaks from August to October. The data on sediment concentration were insufficient to have separate calibration and validation periods.

### 2.3. Model Selection

Since the number of sediment concentration measurement are limited, simulation of sediment contribution from the watersheds requires understanding the local hydrology and the underlying hydrological processes (Steenhuis et al, 2009). The models discussed before (such as SWAT, WEPP etc.) were developed for temperate climates with rainfall throughout the year. The dormant season in temperate climates is related to low temperatures with wet soils at the beginning of the growing season. In monsoon climates, the dormant season is caused by insufficient rainfall and the soils are dry when the rains start. PED (Parameter Efficient Distributed) model (Steenhuis *et al.*, 2009) has been specifically developed for (sub) humid monsoon climates and the WATEM/SEDEM model (Haregeweyn *et al.*, 2013) was designed for the semi-arid monsoon climate.

During the rainy phase in the (sub) humid areas there is more rainfall than that can be evaporated or stored in the soil. The excess water flows downhill as interflow and surface runoff, saturating the bottom lands. Thus a model simulating runoff in the Lake Tana watersheds needs to account for saturation excess runoff (Steenhuis *et al.*, 2009; Tilahun *et al.*, 2011).

The PED model is a physically based runoff and sediment loss model with minimum calibration parameters based on the saturation excess runoff process. It has been applied to catchments ranging from few square kilometers (e.g. Anjeni (1.1 km$^2$, Tilahun *et al*, 2013a), Andit Tid (4.8 km$^2$, Engda, 2011), Enkulal (4 km$^2$, Tilahun *et al,* 2013a) and Debre Mawi (0.95km$^2$, Tilahun *et al.*, 2013b, 2015) to hundreds of thousands square kilometers (e.g. Blue Nile, 180,000 km$^2$) and showed good performance (Steenhuis *et al.*, 2009; Tilahun *et al.*, 2013a; Tilahun *et al.*, 2015). Comparing the predictions with other models used in the humid Blue Nile basin, PED performs as well or better (see Table S1 with model performance statistics in the supplementary material). Hence the PED model is applied in this study to avoid over parameterization and ensure process interpretability.

### 2.4 Description of the PED model

The PED model represents the local hydrological and erosion processes. It classifies the watershed into two runoff producing areas (periodically saturated areas and degraded hill slopes) and one recharge area (permeable hill slopes) that releases the excess precipitation, the base flow and interflow. The two runoff producing areas are assumed to be sources of sediment while the base flow may pick up sediment at low concentrations from the banks. The hydrology model inputs are limited to precipitation, potential evapotranspiration and nine landscape parameters. The sediment model uses the discharges predicted by the hydrology model and maximum six parameters for the erodibility of the soil at the beginning and end of the rainy phase for each of the three areas.

### 2.4.1 PED's hydrology module

The hydrology module is a water balance model that divides the watershed into periodically saturated areas, degraded hill slopes and permeable hill slopes. The discharge Q at the outlet is written as

$$Q = A_1 Q_1 + A_2 Q_2 + A_3 (Q_B + Q_I) \tag{1}$$

where

$A_1$, $A_2$ and $A_3$ are area fractions of the saturated, degraded areas and the recharge hill side area respectively.

$Q_1$ and $Q_2$ are saturation excess runoff from saturated and degraded areas (mm d$^{-1}$),

$Q_B$ and $Q_I$ are base flow and interflow (mm d$^{-1}$) respectively.

Surface runoff is simulated as any rainfall in excess of soil saturation





$$Q_{1,2} = \frac{S_{t-\Delta t} - S_{max} + (P- PET)\,\Delta t}{\Delta t} \qquad \text{when } (P- PET)\,\Delta t > S_{t-\Delta t} - S_{max} \tag{2}$$

where P is precipitation (mm d$^{-1}$), PET is potential evapotranspiration (mm d$^{-1}$), $S_{t-\Delta t}$, previous time step storage (mm) and $\Delta t$ is the time step (d$^{-1}$), $S_{max}$ is the maximum water storage capacity of the three areas;

The storage $S_t$ in each of the three regions is calculated with the Thornthwaite-Mather procedure.

$$S_t = S_{t-\Delta t} + (P - PET)\Delta t \quad \text{when} \quad P \geq PET \tag{3}$$

$$S_t = S_{t-\Delta t} \left[ exp \left( \frac{(P-PET)\Delta t}{S_{max}} \right) \right] \text{when} \quad P < PET \tag{4}$$

Base flow $Q_B$ is calculated as a first order reservoir and interflow $Q_I$ as a zero order reservoir (Steenhuis et al. 2009). The groundwater storage and the recharge to the interflow compartment, $P_{erc, I}$ calculations depend on whether the groundwater storage has reached its maximum value of $BS_{max}$. Recharge to the Interflow only occurs when the baseflow reservoir is full.

The storage in the baseflow aquifer is calculated when the groundwater storage is less than the maximum (i.e., BSt $<BS_{max}$):

$$S_t = BS_{t-\Delta t} + (P_{erc} - Q_B)\Delta t \tag{5a}$$

$$P_{erc,I} = 0 \tag{5b}$$

When the groundwater would exceed the maximum storage (i. e., $BS_{t-\Delta t} + (P_{erc} - Q_B)\Delta t > BS_{max}$):

$$BS_t = BS_{max} \tag{5c}$$

$$P_{erc,I} = BS_{t-\Delta t} + (P_{erc} - Q_B)\Delta t - BS_{max} \tag{5d}$$

The baseflow, $Q_B$ and interflow, $Q_I$, are then obtained as:

$$Q_B = RS_t \frac{[1-exp\,(-\alpha\Delta t)]}{\Delta t} \qquad \text{when BS}_t > BS_{max} \tag{6}$$

$$Q_I = \sum_{\tau=1}^{\tau^*}(2 * P_{erc,I}(\tau^* - \tau)\left(\frac{1}{\tau^*} - \frac{\tau}{\tau^{*2}}\right), \tau \leq \tau^* \tag{7}$$

$P_{erc}$ is percolation to the subsoil (mm d$^{-1}$), $\alpha = 0.69/t_{\frac{1}{2}}$ where $t_{\frac{1}{2}}$ is time taken in days to reduce volume of the base flow reservoir by half under no recharge conditions; $\tau$ is the day after the rainstorm resulting in an amount of percolation, $P_{erc}$ and $P_{erc,I}$ is the amount of the percolate that reached the interflow storage and is calculated as the recharge in excess of what can be stored in the baseflow reservoir and $\tau^*$ is the duration of interflow after any rainstorm.



### 2.4.2 PED's Sediment Module

In the sediment model, the two runoff source areas are considered the main sources of sediment. The seepage from the subsurface flow to the stream channels unlike earlier application of the model (Tilahun *et al.*, 2013a), base and interflow have a small sediment concentration early in the rainy phase (Fox *et al.*, 2007; Fox and Wilson, 2010; Tebebu *et al.*, 2010).

The concentration of sediment, C (g/l): in the river is obtained by dividing the sediment yield by the total watershed predicted discharge from the hydrological model.

$$C = \frac{A_1 Q_1^{1.4}[a_{s,1} + H(a_{t,1} - a_{s,1})] + A_2 Q_2^{1.4}[a_{s,2} + H(a_{t,2} - a_{s,2})] + [(a_{t,3}Q_3^{1.4})]}{A_1 Q_1 + A_2 Q_2 + A_3 Q_3} \qquad (8)$$

where the subscript numbers refer to the three areas introduced with Eq. 1, Q is the runoff (mm/day) calculated with the hydrology model i.e., $Q_1$, $Q_2$ calculated with equation 2, and $Q_3$ is the sum of $Q_B$ in Eq. 6 and $Q_I$ in Eq. 7, H is the fraction of the

contributing runoff area with active rill formation that occurs after plowing and is determined by field observations (Fig. S7) and a is a constant relating the flux to the sediment concentration for each of the three areas with the subscript t for transport limited and subscript s for source limited. Note that unlike in Tilahun *et al.* (2013a) the base flow is not free of sediment in the large river system especially in the beginning of the rainy phase when the sediment is dry and easily picked up.

Measured sediment concentration are required to calibrate parameters, $a_t$ the sediment transport limiting factor and $a_s$, sediment

source limiting factor which are functions of the slope, Manning's roughness coefficient, slope length, effective deposition and vegetation cover (Yu *et al.*, 1997). In the calibration flow used in rating curve development are compared to predicted flow and in a few cases when it was greatly different, we took the flow on the day before or after when they were more similar to the observed flows.

### 2.5 Model calibration, validation and setup

All the model parameters were calibrated on daily basis for 1994-1999 and validated for 2000 2009. The parameters are first determined by maximizing the efficiency criterion of the Nash–Sutcliffe efficiency coefficient (NSE) then the coefficient of determination ($R^2$) and finally minimizing the Root Mean Square Error (RMSE). For calibration of parameters of the hydrology model, we started by giving initial values of three physical area model parameters $A_1$, $A_2$ and $A_3$ and the maximum storage process parameters $S_{max}$ of the three areas and sub-surface parameters ($BS_{max}$, $\tau^*$ and $t_{1/2}$ (half-life)). The

initial values were based on the previous model runs of Steenhuis *et al.* (2009) and Tilahun *et al.* (2013a). These initial values were changed systematically until the best goodness-of-fit was achieved between simulated and observed flows.

Daily sediment concentrations were computed by calculating daily sediment load first and then dividing the daily load by the total daily discharge. In the sediment model, there are two calibration parameters for each of the two surface runoff source areas $A_1$ and $A_2$ for transport limit $a_t$, at the beginning of the rainy phase, and source limit, $a_s$, at the end of the rainy phase



and for the interflow and base flow (A₃) that represents the sediment that is being picked up in the river channel during low flows. These constants are tweaked to yield a best fit between measured and simulated daily sediment concentrations.

## 3 Results and discussion

### 3.1 Hydrology

#### 3.1.1 Evaluation of the hydrology module

The model parameters are shown in Table 2 and show a reasonable agreement (Table 3)for all four basins except for the portion of the hillside (A₃) in which the water infiltrates that supplies water for the interflow and the base flow that is smaller for the Megech and Rib than for the Gilgel Abay and the Gumara. Previous sensitivity analysis has shown that the relative areas, aquifer half-life and the duration of the interflow after a rainstorm are the most sensitive parameters (Tilahun et al, 2013, a, b). Especially the maximum storage, $S_{max}$, can be changed over a wide range before it affects the outflow predictions.

**Gilgel Abay**: Using a runoff contributing area of 15 % ( 5% saturated and 10% degraded), an aquifer half-life of 45 day and a 40-day interflow period (Table 2), the predicted and observed daily discharge for the Gilgel Abay shows good agreement for the calibration period from 1994 to 1999 (hydrograph in Fig. 2A NSE= 0.77, Table 3) . The fluctuations during the high-flow periods in some of the years were not captured which in part is caused by our inability to estimate amounts or rainfall accurately due to the sparse rain gauge network (Dessie et al. 2014). There was also an anomaly in the collected data in the fall of 1996. During the validation (2000-2009) the rising and falling limbs and most of the peaks were reasonable well estimated (Fig. 2B; NSE= 0.71, Table 3). The Pbias values of 2.62 (Table 3) for calibration indicate also that the model performed well (Pbias value for flow ±25 is acceptable, Moriasi *et al.*, 2007) but slightly underestimated the flow initially and then minimally overestimated later on. The base flow after 2006 that increased unexpectedly compared to the previous years was underestimated, suggesting a slight change in channel configuration, as discussed later, that was not reflected in the rating curve.

**Gumara:** With nearly the same parameter set as for the Gilgel Abay model (Table 2) the model performed generally well to predict the discharge. The NSE values for daily flows were 0.70 for calibration and 0.77 for validation period (Fig. 3, Table 3). A smaller number of missing precipitation data during validation likely was related to better model performance.

**Rib:** Although the surface runoff parameters for areas A₁ and A₂ used in the model for the Rib are similar to both the Gumara and the Gilgel Abay, the subsurface parameters are much different (Table 2). The area contributing to the stream flow, the half-life and duration of the interflow period are all significantly less than for the Gumara and Gilgel Abay (Table 2). Despite that the daily flows were predicted reasonably well for calibration period with NSE values for daily discharges of 0.71. In the calibration period the daily NSE decreased to 0.55 (Table 3). In the calibration and especially validation period

the peak values were over predicted because of the observed discharges are limited to bank full discharges which are around 6 mm/day as is discussed below. During the period of September to January in 1996 and 1997, more base flow was observed similar to the Gilgel Abay (Fig. 4A).

**Megech:** The Megech river has a reservoir upstream and flow is attenuated therefore the flow is summarized in 10 days to

avoid the effect of the reservoir. The $R^2$ is 0.85 and 0.79 and NSE is 0.71 for calibration and 0.31 for validation on a ten day basis (Table 3, Fig. 5). The latter is caused by an unexpected and unlikely reduction in observed flows starting 2006.

**Discussion of discharge predictions**: It is remarkable that the surface flow parameters for all four watersheds are nearly the same especially if we take the relative insensitivity of the $S_{max}$ value in determining the simulated discharge into account. These values are also similar compared with other watersheds where we used PED (Tilahun 2013a, b).  Moreover, it is

curious that starting with the same year 2006 the observed discharge values are less for the predicted base flow of the Gilgel Abay and all discharge predictions of the Megech prediction (Fig 2 and 5). This might be explained by the occurrence of the large amounts of rainfall during that year which affected the river bed and consequently the water table levels and calculated discharges with the rating curves that were not recalibrated (SMEC, 2008).

The fractional areas for Gilgel Abay and Gumara add up to 1 but Rib and Megech are only 0.6 and 0.65 (Table 2). An area

proportion of one means the calculated interflow, base flow and storm flow are equal to the long term discharge measured at the outlet. In other words since the long term average of the discharge in the PED model equals the average of net precipitation (i.e., rainfall minus evaporation), all precipitation reaches the outlet eventually. However, the total contributing areas of  0.6 for Rib and 0.65 for Megech means that the net input precipitation is much more than the discharge at the outlet. Thus the unaccounted net precipitation either flows subsurface under the gauge to the lake or the discharge is not measured

correctly.

The under prediction of the high flows for the Rib river is a consequence of the increased bed levels  (SMEC, 2008), can be observed clearly in Fig. 4A in our paper and in a recently published manuscript by Dessie et al (2014). In Fig. 4a the under prediction is indicated with ellipses 1 and 2 in which the measured flows do not exceed an equivalent of 6 mm/day with predicted flows much greater than that. Thus after the river reaches a level of flow equivalent to 6mm/day the river's banks

overflow and is not represented by the rating curve. The under prediction of the high flows at the Rib stream gauge is even more clear in Dessie *et al.* (2014) where in the period from July 10 to September 15, 2012 a newly installed upstream gauge of the Rib shows the weekly peak flows are up to 300 $m^3$/sec (equivalent of 22 mm/day) but in the downstream gauge the peak flows are invariably at 150 $m^3$/sec (equivalent to 10 mm/day).  Since rivers are extremely flashy the peak runoff occurs only part of the day and the 6 mm/day observed over the whole day (this paper) is comparable with the 10 mm/day over part

of the day (Dessie *et al.*, 2014). The final cause for the "missing" rainfall (i.e., contributing areas not adding up to 1 is that

the Rib watershed is underlain by permeable tuffs (Dessie *et al.*, 2014) facilitating sub surface flows and this decreases the amount of the flow at the gauge.

Under prediction of the flow by the Megech River (Figure 5) is either due to Angereb Dam or since the area is volcanic subsurface flow under the gauge could occur as well. The over prediction of the base flow, after 2006, is likely caused by a
change in the riverbed as discussed above.

### 3.2 Sediment

### 3.2.1 Evaluation of the sediment module

The sediment concentrations were measured by the Ministry of Water, Irrigation and Electricity (MoWIE) as part of determining the sediment rating curves for each of the four rivers in the Lake Tana basin. The measured sediment
concentrations (Fig. 6) show as expected that large flow events are related with high concentrations. In addition, for similar runoff events the concentrations are greater during the onset of rainfall phase than later on.

The concentration measurements were employed to calibrate the "a" parameters (Eq. 8) in the sediment module. The surface runoff for the 16 years originating from the saturated and degraded area and the subsurface flow (interflow and base flow) that are required for the sediment module were predicted with the PED hydrology module.  The trend of decreasing sediment
concentration is captured by the H function (Eq. 8) with cumulative rainfall (Fig. S7 in the supplementary material)

The predicted and observed sediment concentrations as a function of time for Gumara for which we have the most of the sediment data points are shown in Fig. 7 with the input parameter set in Table 4.  Sediment concentrations for the other watersheds with fewer observations are shown in the supplementary material as Figure S4-S6 for the Gilgel Abay, Rib and Megech.   Daily observed vs corresponding predicted sediment concentrations for the four watersheds agree well as shown in
(Table 5) with Nash Sutcliff values ranging from 0.5 to 0.84.

The average sediment loads from these watersheds (1994-2009) are 35 t/ha/yr for Gilgel Abay, 49 t/ha/yr for the Gumara, 25 t/ha/yr for the Rib and 12 t/ha/yr for the Megech (Table 7a). The Gumara transport most sediment because the losses per unit area are almost 49 ton/ha and is greater than any other river. The load of the Gilgel Abay is elevated because runoff losses are much greater than any other watershed.

The concentration in the water when the rill is formed is related to the maximum amount of sediments that can be carried by the water in the rill (Zegeye *et al.*, 2010; Tebebu *et al.*, 2010) and is represented in the $a_t$ coefficient in the model. Its magnitude is related to the stream power which is a function of the slope of the land (Gao, 2008).  Since the slopes in the Gilgel Abay watershed are relatively flatter than the Rib and Gumara (Fig. S1, Table 6), the transport capacity for the Gilgel Abay in Table 4 is less than for the Rib and the Gumara.  The Megech has reservoir upstream of the sampling location that
takes out most of the sediment and can explain, therefore, the low transport coefficient despite the steep terrain (Table 4).




Once the rill network is formed, it is reasonable that sediment concentration decrease and at the end of the rainfall monsoon phase the concentration is source limited and depends on the cohesion of the soil and depends on the soil type. This is represented by the source limit term, $a_s$, in the model. The soils in the Gumara and Rib have a greater percentage of chromic Luvisols (loamy sand, Fig. S2) than in the Gilgel Abay and Rib. Therefore, the $a_s$ values in Table 4 are less for the Gumara and Rib than for the Gilgel Abay.

### 3.3 Sediment contributions to Lake Tana

Table 7a shows the amount of sediment for the four rivers of the gauged part of the basin (1994-2009) equal to 16 Tg/yr (Tg is equal to $10^{12}$ grams or $10^6$ tons). In table 7b the sediment budget is detailed for Lake Tana and the floodplains. The floodplains are found near the lake (Figure 1) and act as storage for flood (Dessie *et al.*, 2014). During storage sediment settles on the land and the floodplains are the sinks for sediment. SMEC (2008) found that an area of 350-450 km$^2$ is inundated around Lake Tana during floods. This inundated floodplain area is approximately 6% of the ungauged part (or 4% of the whole watershed)

For the purpose of the sediment budget, the floodplain includes the deltas that have been formed at the mouth of the river in the lake. Assuming as discussed above that the flood plain and the lake are sediment sinks and the remaining part of the watershed are sediment sources, the sediment budget for the floodplain can then be written as:

$$S_{flplane} = M_{gauged} + M_{ungauged} - S_{lake} - M_{blNile} \qquad (9)$$

where $S_{flplane}$ is the amount of sediment stored in the floodplain, $M_{gauged}$ is the annual sediment loss from the gauged part of the basin, $M_{ungauged}$ is the sediment loss from the ungauged part of the basin, $S_{lake}$ is the annual amount of sediment stored in the lake and $M_{blNile}$ is the sediment lost per year at the outlet of the lake in the Blue Nile. Consequently, the difference between the incoming sediment from the watershed and that leaving the lake is deposited in the lake and in the floodplains. We will discuss now each of the term in the sediment balance equation.

Based on an analysis by Ayana (2007) who compared the bathymetric surveys of Lake Tana in 1987 and 2006 found that 200 Mm$^3$ of sediment had settled at the bottom of the lake, the annual sediment load deposited in the Lake, $S_{lake}$ can be calculated assuming a bulk density of 1,200 kg/m$^3$ as 12 Tg/yr (trillion of grams per year or million tons per year, Table 7b).

The 1.6 Tg/yr of sediment leaving the lake in the Blue Nile (Table 7b) was found by multiplying the monthly average of the available measured concentrations at the outlet , (Table S1 in the supplementary material) by the monthly average discharge. The sediment delivered to the lake is the sum of what is settled at the lake bottom plus that has left the lake and equals 13.6 Tg/yr (Table 7b).

In order to estimate the total amount of sediment that can be contributed from the uplands to the lake, we need to bracket the amount of sediment coming from the ungauged part. The lower bound (although not very likely) is when the ungauged part does not contribute sediment. The upper bound of the soil loss can be found by noting that the distribution of slopes in the ungauged part of the basin is very much the same as the Gilgel Abay especially if we subtract the 6 % for the 0-2% slope

class (Table 6). Thus a reasonable estimate for the upper bound of the soil loss for the ungauged part can be obtained by using the sediment parameters for the Gilgel Abay for the ungauged part of the watersheds assuming that the rainfall and runoff remains the same as for the gauged part of the particular watershed. The results of these calculations are given in Table 7a. We find that the total loss from the ungauged parts of the four rivers is 11.3 Tg/yr or an average soil loss of 31.5 tons per ha (Table 7b).

There are additional areas in the Lake Tana basin that are not part of the four large watersheds. We will assume that these areas have the same soil loss as the average soil loss per ha as of the ungauged watersheds or 31.5 ton/ha. Thus by multiplying the total acreage of ungauged basin (minus the floodplains) with the average soil loss of 31.5 ton/ha, the upper bound for sediment contribution of all the ungauged part of the basin outside the floodplain is 21.5 Tg/yr (Table 7b).

The final sediment budget calculations and the portion of the sediment retained in the floodplain, the lake and both the flood

plain and lake together are shown Table 7c for both the lower and upper bounds of the sediment contributed by the ungauged basins. The portion retained can be calculated simply as the sediment retained divided by the incoming sediment.

The amount of sediment retained in the flood plain is greatly dependent on the amount of sediment delivered from the gauged and ungauged parts. Fifteen percent of the sediment is retained in the floodplain for the lower bound for the unlikely scenario that the there is no sediment generated in the ungauged part of the watershed. For the scenario that the ungauged

basin has the same characteristics as the Gilgel Abay we find as upper bound that 64% is retained in the floodplain (and deltas at the river mouth).

The annual sediment load that comes into the lake is equal to the sediment deposited in the lake 12 Tg/year plus the amount leaving the lake (1.6 Tg/yr). Since these two quantities are measured it is independent of predicted amount of sediment originating from the watershed and it is equal to 88%. Finally the sediment retained in both the flood plain and lake varies

between 82% (lower bound) and 96% (upper bound, Table 7c).

The evidence of the near shore deposition is most obvious for the Gilgel Abay (with a relatively small floodplain) that has formed a peninsula of around 10 kilometers long and 2 km wide. Gumara and Rib rivers have a large flood plain area and the additional land formed offshore which is around 0.8 by 3 km (Abate *et al.*, 2015) is comparatively smaller.

In addition our prediction of the PED model can be compared with a water supply reservoir (Angereb) in the Lake Tana

basin that has accumulated 1.8 M m$^3$ of sediment within 11 years (Haregeweyn *et al.*, 2012). The sediment accumulated

within the reservoir is 0.2 Tg/year for the 68 km$^2$ watershed. This is equivalent to 29 t/ha/yr. This value is similar order of magnitude to the sediment delivered by other watersheds.

### 3.4 Implications

Based on our simulation and understanding of the local hydrology; sources of sediment are degraded and saturated bottom lands. According to our model most sediment are generated from the degraded areas (166, 153, 128 and 50 km$^2$ areas in Gilgel Abay, Gumara, Rib and Megech respectively, Table 2). The soil that is lost from the saturated areas is likely from the gullies that are found in the periodically saturated bottom lands (Bayabil *et al.*, 2010; Tebebu *et al.*, 2010; Tilahun *et al.*, 2013a) .Therefore implementation of best management practices should be targeted on degraded areas and saturated bottom lands and especially on  gully rehabilitation.

Finally it is of interest for future sediment contributions to the lake that during the simulation period the only reservoir functioning in the watershed was the Angereb watershed.  The Koga reservoir started to store water in 2009. To date the construction of Megech and Rib reservoirs is in progress and reservoirs on Gumara and Gilgel Abay are under study. Sediment will be trapped in these reservoirs.  How much this will benefit the lake is not obvious since flooding will decrease as well and thus settling of the soil on the flood plain.

### 4 Conclusions

The sediment contribution of major watersheds within Lake Tana basin was estimated. On average annual sediment loss from the gauged part of the watershed is 32 t/ha. Accumulation of sediment in Lake Tana is taking place at an annual rate of 10 t/ha of watershed area with a small amount of outflow from the lake to the Blue Nile River.  The main problem in calculating the amount of sediment retained in the floodplain and within the lake is to estimate the soil loss from the 60% of the basin that was not gauged. Therefore we established upper and lower bounds. The upper bound assumed that the sediment loss for the ungauged part was equivalent per unit area to the Gilgel Abay which had a similar landscape.  The lower bound was zero which is unrealistic. Based on these estimates we estimated that the amount of sediment retained by the lake is 88% of what enters into the lake and that retained in the near shore areas and in the flood plains (mainly) is between 15 and 64% of the sediment generated in the uplands. The sediment retained by the floodplain and lake is between 82 and 96%.

Priorities on regional implementation of soil and water conservation should be based on the rate of degradation. The basin with the greatest sediment transport is the Gumara followed by the, Gilgel Abay, Rib, and Megech. Despite its smallest loss the Megech might need treatment too since the reservoir is being filled up.




Flood plains play an important role in trapping sediment. However when the proposed dams are functional and flooding is controlled, less of the sediment amount will be taken out of the water. Sediment rich water from rivers not flowing in the reservoir will flow inside the existing channels to the lake without flooding the plains.

5  *Acknowledgements* Funding for this research was obtained from the USAID through the research project called "Participatory Enhanced Engagement in Research" or PEER Science project (grant number AID-OAA-A-11-00012) Additional funding was also obtained from Higher Education for Development (HED), United States Department of Agriculture (USDA) and funds provided by Cornell University partly through the highly appreciated gift of an anonymous donor.



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





Table 1: Model performance of PED and other models

| | PED (Steenhuis et al,2009) BNB* | | | | SWAT(Betrie et al,2011) BNB | | SWAT(Easton et al, 2010) BNB | | | | PED (Tilahun et al,2013a, Tilahun et al, 2013b) Anjeni / Debre Mawi | | | |
|---|---|---|---|---|---|---|---|---|---|---|---|---|---|---|
| | $R^2$ | | NSE | | NSE | | $R^2$ | | NSE | | $R^2$ | | NSE | |
| | Cal | Val | Cal | Val | Cal | Val | Cal | Val | Cal | Val | Cal | Val | Cal | Val |
| Flow | 0.98[2] | 0.8[2] | 0.98[2] | 0.73[2] | 0.68[1] | 0.63[1] | 0.92[1] | - | 0.87[1] | - | 0.88[1] /0.79[1] | 0.82[1] | 0.86[1] /0.82[1] | 0.80[1] |
| Sed. | 0.81[2] | 0.74[2] | 0.75[2] | 0.69[2] | 0.88[1] | 0.83[1] | 0.67[4] | - | 0.67[4] | - | 0.8[1] /0.74[1] | 0.67[1] | 0.78[1] /0.8[1] | 0.64[1] |

*BNB: Blue Nile Basin

5    1-Daily, 2-10 day averaged,3-Monthly,4-seasonal





Table 2: PED Parameter values of hydrological and sediment concentration for four major rivers in the Tana Basin

| Parameters | Unit | G_Abay | Gumara | Rib | Megech |
|---|---|---|---|---|---|
| Area $A_1$ | | 0.05 | 0.05 | 0.05 | 0.05 |
| $S_{max}$ in $A_1$ | mm | 65 | 90 | 100 | 100 |
| Area $A_2$ | | 0.1 | 0.12 | 0.1 | 0.1 |
| $S_{max}$ in $A_2$ | mm | 35 | 40 | 30 | 30 |
| Area $A_3$ | | 0.85 | 0.83 | 0.45 | 0.5 |
| $S_{max}$ in $A_3$ | mm | 125 | 100 | 135 | 250 |
| $BS_{max}$ | mm | 70 | 75 | 75 | 80 |
| $t_{1/2}$ | days | 45 | 50 | 20 | 30 |
| $\tau*$ | days | 40 | 40 | 25 | 20 |

5  $A_1$, $A_2$ and $A_3$ are area fractions of the saturated, degraded and recharge hillside areas respectively. $S_{max}$ is the maximum water storage capacity; $BS_{max}$ is maximum base flow storage of linear reservoir; $t_{1/2}$ is time taken in days to reduce volume of the base flow reservoir by half under no recharge conditions; $\tau*$ is the duration of the period after a single rainstorm (until interflow ceases).




Table 3: Efficiency criteria for calibration and validation of discharge in mm/day for the four major rivers in the Tana Basin

| Watersheds | Description | | Calibration | | Validation | |
|---|---|---|---|---|---|---|
| | | | Daily | Monthly | Daily | Monthly |
| Gilgel_Abay | Mean | Predicted | 2.9 | 87.1 | 3.0 | 92.0 |
| | | Observed | 2.8 | 89.5 | 2.8 | 84.9 |
| | | $R^2$ | 0.77 | 0.91 | 0.75 | 0.94 |
| | | NSE | 0.77 | 0.91 | 0.71 | 0.87 |
| | | RMSE | 1.8 | 32.0 | 1.9 | 35.0 |
| | | RVE | -0.01 | 0.03 | -0.09 | -0.09 |
| | | Pbias | 2.62 | 2.61 | -8.42 | -8.42 |
| Gumara | Mean | Predicted | 2.4 | 74..7 | 2.2 | 66.8 |
| | | Observed | 2.6 | 78.6 | 2.5 | 77.5 |
| | | $R^2$ | 0.72 | 0.87 | 0.78 | 0.92 |
| | | NSE | 0.70 | 0.86 | 0.77 | 0.90 |
| | | RMSE | 2.12 | 40.33 | 1.99 | 36.27 |
| | | RVE | 0.04 | 0.05 | 0.14 | 0.14 |
| | | Pbias | 4.9 | 13.8 | 4.9 | 6.5 |
| Rib | Mean | Predicted | 1.1 | 31.8 | 1.0 | 30.5 |
| | | Observed | 1.m1 | 32.8 | 1.0 | 31.7 |
| | | $R^2$ | 0.72 | 0.91 | 0.64 | 0.84 |
| | | NSE | 0.71 | 0.90 | 0.55 | 0.81 |
| | | RMSE | 1.02 | 15.75 | 1.12 | 19.84 |
| | | RVE | 0.01 | 0.03 | -0.03 | -0.03 |
| | | Pbias | 2.9 | 2.9 | 3.9 | 1.5 |
| ......Megech | Mean | Predicted | 12.0 | 34.5 | 13.2 | 30.2 |
| | | Observed | 11.5 | 36 | 10.1 | 39.8 |
| | Decadal | $R^2$ | 0.85 | 0.91 | 0.79 | 0.84 |
| | | NSE | 0.71 | 0.76 | 0.31 | 0.66 |
| | | RMSE | 11.4 | 24 | 13.1 | 34.1 |
| | | RVE | 0.04 | -0.04 | 0.24 | -0.32 |
| | | Pbias | -4.4 | -4.4 | -31.3 | -31.8 |





Table 4: Simulated sediment concentration parameters for the transport limit, $a_t$ and the source limit, $a_s$ in $((g/l)(mm/day)^{0.4})$ for the four main rivers in the Lake Tana basin

| Source | G_Abay | | Gumara | | Rib | | Megech | |
|---|---|---|---|---|---|---|---|---|
| | $a_t$ | $a_s$ | $a_t$ | $a_s$ | $a_t$ | $a_s$ | $a_t$ | $a_s$ |
| Saturated | 3 | 2.5 | 7 | 4 | 8 | 5 | 2.5 | 1.5 |
| Degraded | 5 | 5 | 15 | 5 | 10 | 5 | 5 | 4.5 |
| River | 0.7 | 0 | 0.8 | 0 | 0.6 | 0 | 0.15 | 0 |



Table 5: Efficiency criteria for simulated vs observed sediment concentrations

| 1994-2009 | | Daily sediment concentration (g/l) | | | |
|---|---|---|---|---|---|
| | | Gilgel Abay | Gumara | Rib | Megech |
| Mean | predicted | 1.55 | 3.32 | 4.76 | 0.84 |
| | observed | 1.70 | 3.24 | 4.60 | 0.79 |
| | $R^2$ | 0.67 | 0.56 | 0.7 | 0.84 |
| | NSE | 0.65 | 0.67 | 0.73 | 0.84 |
| | RMSE | 0.84 | 1.25 | 1.71 | 0.38 |
| | Error | 0.25 | -0.08 | -0.16 | 0.02 |





Table 6 Slopes in the gauged watersheds (Gilgel Abay, Gumara, Rib, Megech) in the Lake Tana basin and the remainder of the lake Tana basin not gauged

| Slope (%) | G_Abay | Gumara | Rib | Megech | Ungauged basin |
|---|---|---|---|---|---|
| 0-2 | 22 | 7 | 7 | 1 | 29 |
| 2-8 | 50 | 25 | 30 | 17 | 42 |
| 8-16 | 19 | 31 | 25 | 26 | 15 |
| 16-30 | 9 | 28 | 22 | 32 | 8 |
| >30 | 1 | 9 | 16 | 25 | 5 |



Table 7a: Sediment budget for gauged and ungauged parts of the four Lake Tana watersheds: Gilgel Abay, Gumara, Rib, Megech

| Watersheds | Gauged | | | Ungauged | | | Total for four rivers Tg/yr |
|---|---|---|---|---|---|---|---|
| | Area (km²) | Total Tg/year[1] | Unit area Mg/ha/yr | Area (km²) | Total Tg/yr | Unit area Mg/ha/yr | |
| Megech | 500 | 0.6 | 12.2 | 163 | 0.3 | 21.0 | 0.9 |
| Gumara | 1281 | 6.3 | 49.4 | 688 | 1.9 | 28.0 | 8.2 |
| Rib | 1289 | 3.2 | 24.6 | 379 | 0.7 | 17.7 | 3.9 |
| G_Abay | 1665 | 5.9 | 35.4 | 2362 | 8.4 | 35.4 | 14.3 |
| Total | 4735 | 16.0 | | 3592 | 11.3 | | 27.3 |
| Average | | | 33.8 | | | 31.5 | 32.8 |

[1]$10^{12}$ grams or 1 million tons

Table 7b: Sediment budget for the Lake Tana consisting of predicted minimum and maximum contribution from the watershed and the measured sediments deposited in the lake and that leaving the Lake at the outlet. The minimum contribution assumes that the ungauged part of the Lake Tana watershed does not contribute sediment and the maximum contribution assumes that the ungauged part has similar landscape characteristics as the Gilgel Abay

| | Area (km²) | Minimum contribution | | Maximum contribution | |
|---|---|---|---|---|---|
| | | Mg/ha/yr | Tg/yr | Mg/ha/yr | Tg/yr |
| **Watershed contribution** | | | | | |
| Gauged river contributions (predicted, Table 7a ) | 4735 | 33.8 | 16.0 | 33.8 | 16.0 |
| Ungauged river contributions minus floodplains(estimated, table 7a ) | 6829 | 0 | 0 | 31.5 | 21.5 |
| Total watershed contribution minus floodplains | 11564 | 13.8 | 16.0 | 32.4 | 37.5 |
| **Sediment reaching the lake** | | | | | |
| Deposited in the lake (measured) | 3000 | | 12.0 | | 12.0 |
| Outflow from the lake (measured) | | | 1.6 | | 1.6 |
| Total sediment reaching the Lake (measured) | | | 13.6 | | 13.6 |
| Retained in floodplains and deltas (calculated) | 436 | 55 | 2.4 | 548 | 23.9 |

Table 7c: Annual sediment mass balance of Lake Tana and its floodplain

| | Lower bound | | | Upper bound | | |
|---|---|---|---|---|---|---|
| | Delivered Tg/year | Retained | | Delivered Tg/year | Retained | |
| | | Tg/year | Portion | | Tg/year | Portion |
| Floodplain and deltas (Table 7b) | 16.0 | 2.4 | 0.15 | 37.5 | 23.9 | 0.64 |
| Lake( Table 7b, measured) | 13.6 | 12 | 0.88 | 13.6 | 12 | 0.88 |
| Floodplain, deltas and lake | 16.0 | 14.4 | 0.90 | 37.5 | 35.9 | 0.96 |



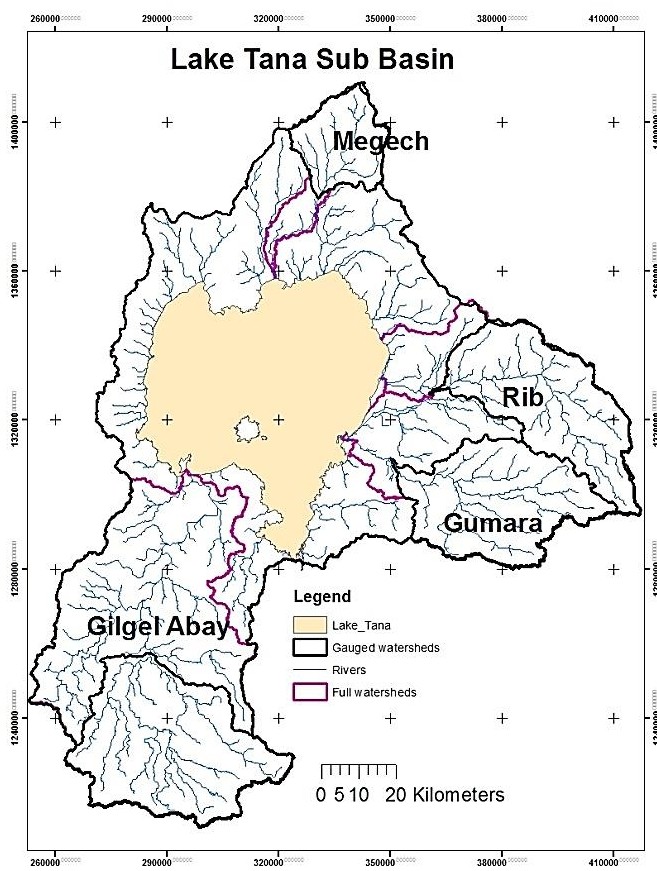

Figure 1: The Lake Tana basin containing four major watersheds: Gilgel Abay, Gumara, Rib and Megech





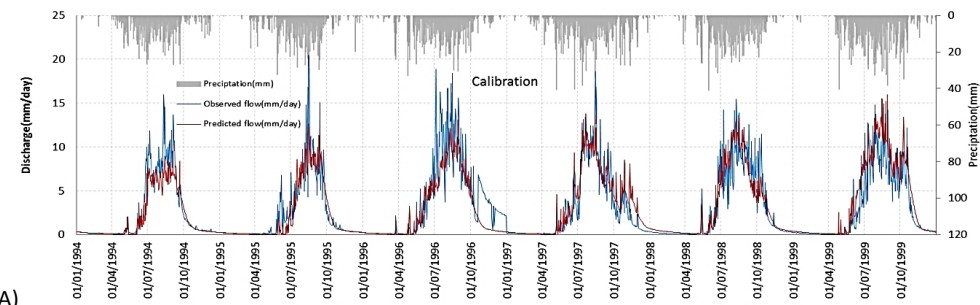

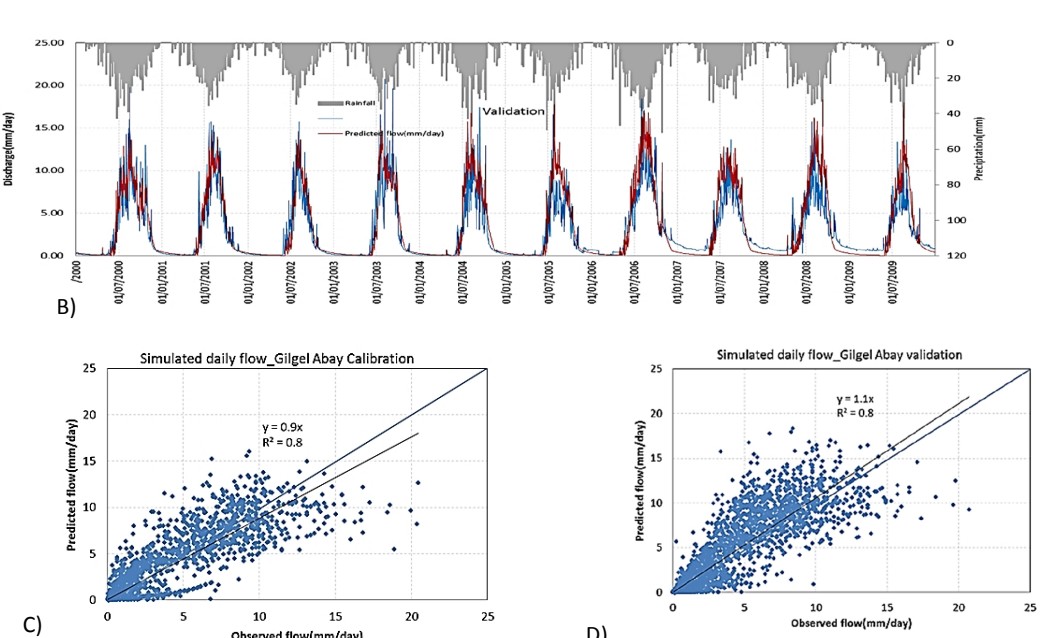

Figure 2: Simulated and observed daily stream flows in mm/d for Gilgel Abay River. A) Calibration (1994-1999), B) Validation(2000-2009), C) Scatter plot of simulated vs observed for Calibration, D) Scatter plot of simulated vs observed validation



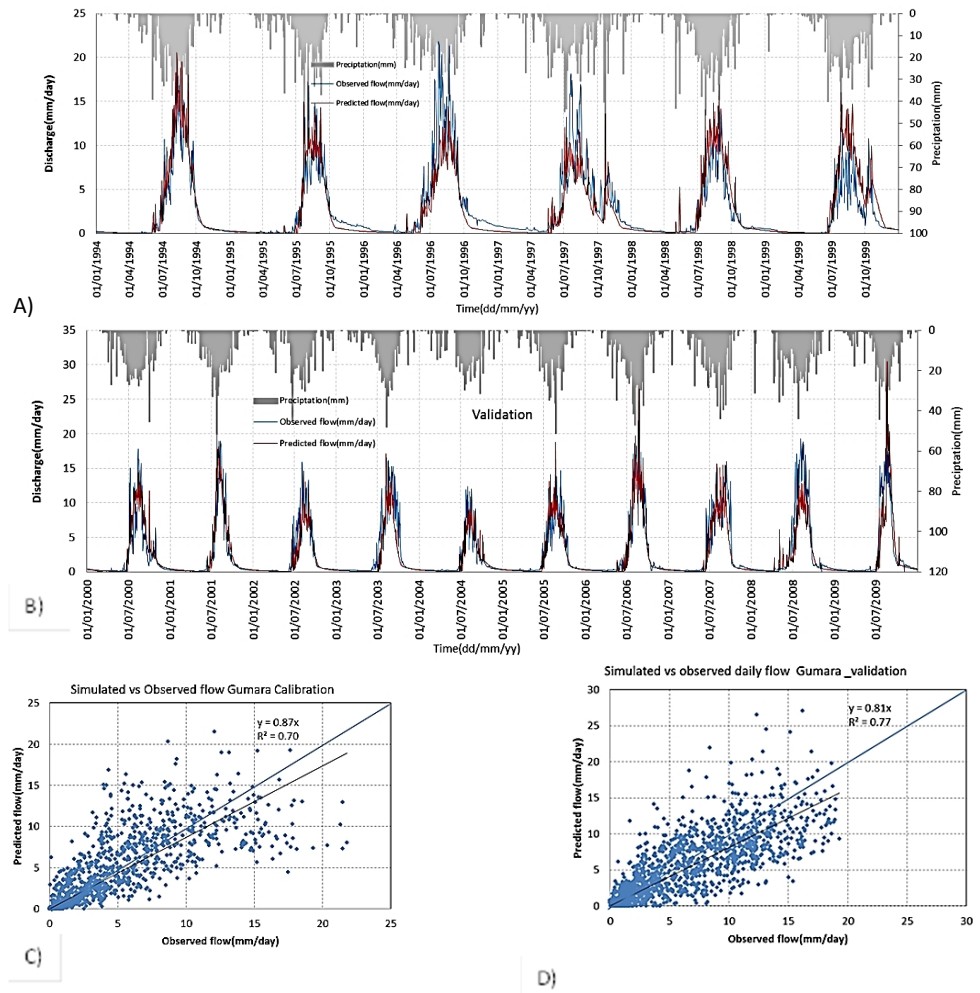

Figure 3: Simulated and observed daily stream flows in mm/d for Gumara River. A) Calibration (1994-
1999), B) Validation(2000-2009), C) Scatter plot of simulated vs observed for calibration, D) Scatter
plot of simulated vs observed validation



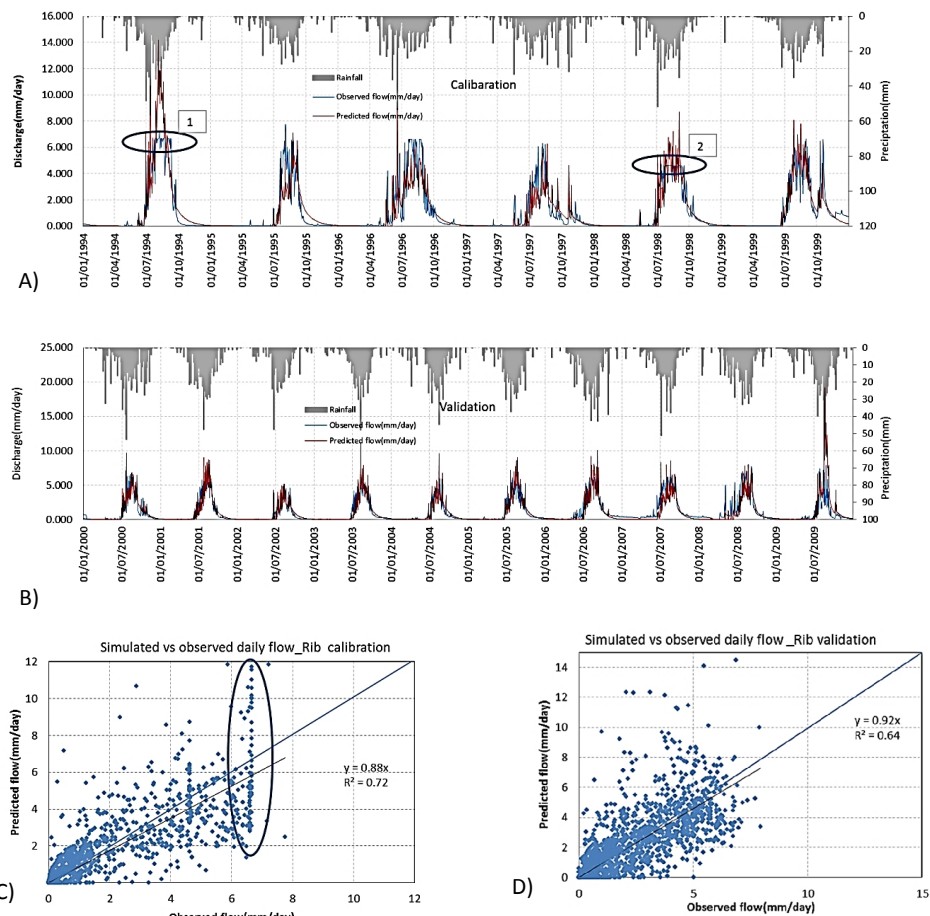

Figure 4: Simulated and observed daily stream flows in mm/d for Rib River. A) Calibration (1994-1999), B) Validation(2000-2009), C) Scatter plot of simulated vs observed for Calibration, D) Scatter plot of simulated vs observed validation





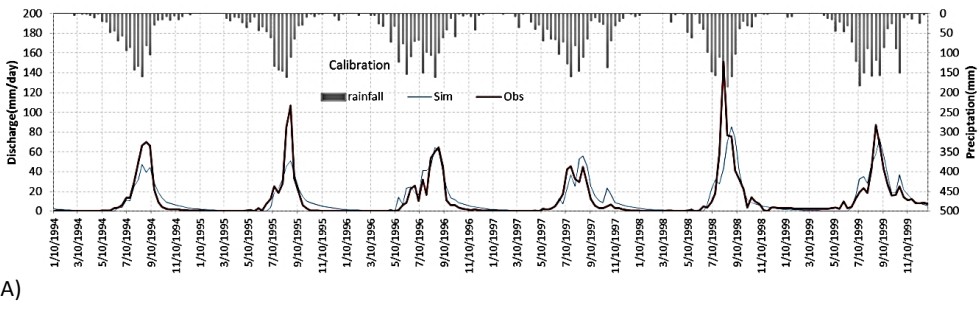

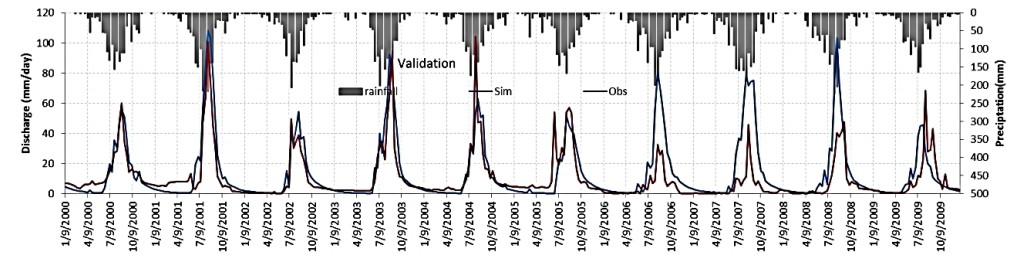

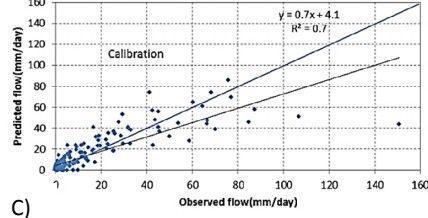

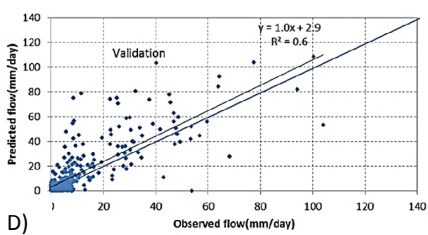

Figure 5: Simulated and observed averaged 10 day stream flows in mm/d for Megech River. A) Calibration (1994-1999), B) Validation (2000-2009), C) Scatter plot of simulated vs observed for calibration, D) Scatter plot of simulated vs observed for validation


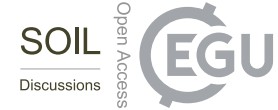

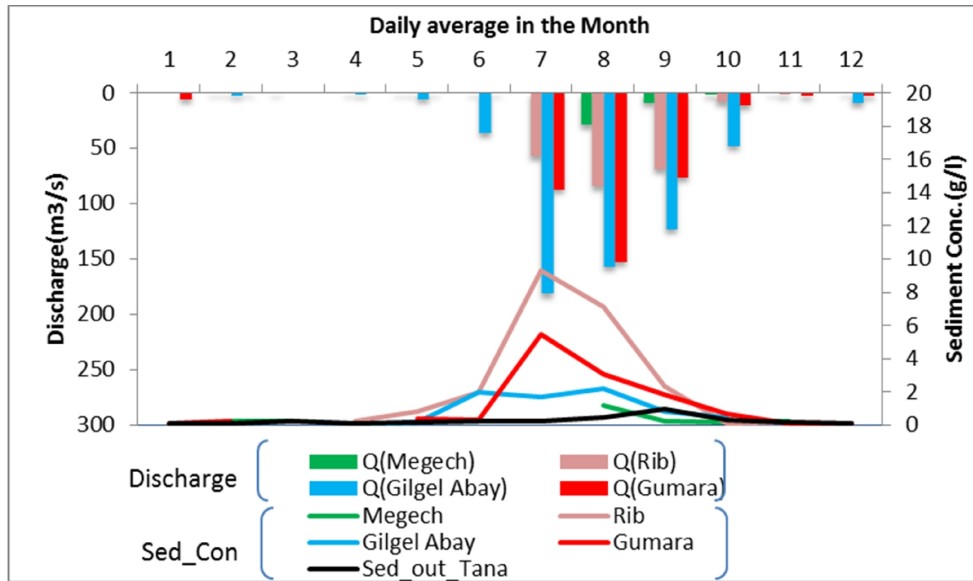

Figure 6: Average daily sediment concentration (1964-2009) (g/l) and discharge (m3/s) of the four major rivers in the Tana Basin





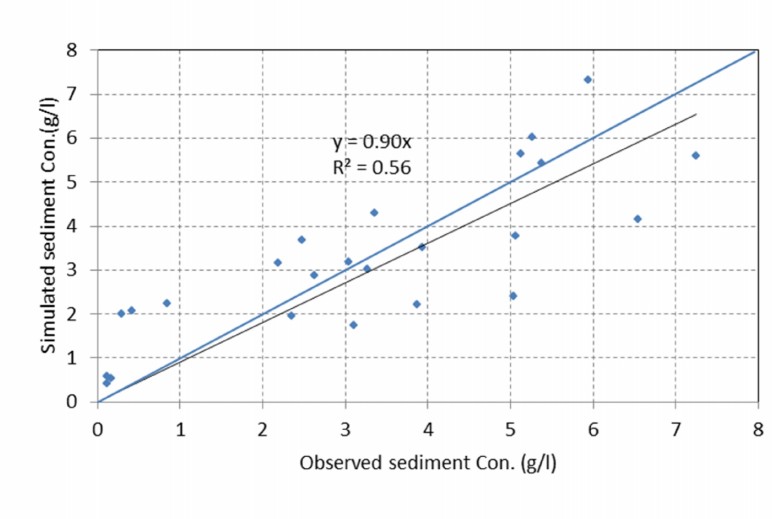

Figure 7: Simulated vs Observed sediment concentration for Gumara watershed

