# Peer review of "Calculating the sediment budget of a tropical lake in the Blue Nile basin: Lake Tana"

_SOIL, 2015_

## Referee Comment (RC1) · Anonymous Referee #1 · 11 Feb 2016

Review SOIL-2015-84: Calculating the sediment budget of a tropical lake in the Blue Nile basin: Lake Tana

The authors present the calculation of the past sediment budget of Lake Tana, Ethiopia, that is threatened by pollution by sediments and nutrients. The calculation is based on already existing data as well as on discharge and sediment modelling with the PED model. The topic of the paper is of high environmental interest, however the scientific and presentation quality are only moderate to poor.

I recommend *rejection* for the following reasons:

*General comments:*

1. The objectives in the Introduction need to be put clearer. A hypothesis was not formulated.
2. It seems that measured precipitation, discharge and sediment data were not analyzed for consistency sufficiently. Measured data that are proven to be not correct need to be excluded or corrected before further analysis and cannot be used for model calibration and validation. The authors should redo the analyzation of the used data. After that the results might be different from the results presented here.
3. The manuscript needs a major restructuring. Methods and results need to be strictly differentiated.
4. The description of some methods and used data is missing (see Specific comments for details) so that some argumentation cannot be understood.
5. The results should be discussed in more detail and should not be related to errors/anomalies in the measured data.
6. Figures and tables in the supplementary material are either superfluous or could be integrated directly in the manuscript (see Specific comments for details).

*Specific comments:*

*Text:*

- P. 2, l. 8ff: Give the size of the watersheds in brackets.
- P. 3, l. 17ff: The source of the meteorological data needs to be given. How many stations are used? Give the coordinates and names. How long is the measurement period? What is the temporal resolution of the measurements?
- Parts 2.1.1 to 2.1.4: Percentage values should be given either as numbers or as text but should not be intermixed. Instead of references to figure as 1 a reference to table 6 should be given.
- P. 4, l. 2: Why do you call it Lake Tana "sub" basin and not simply basin?
- P. 5, l. 1ff: See comment before: Which stations are used? How long is the measurement period? What is the temporal resolution of the measurements? If there are several stations, why were missing values not estimated by correlations with the surrounding stations? I guess this would better represent the temporal variability of the meteorological data.

- P. 5, l. 6ff: Give the names and coordinates of the gauges. Describe in more detail how the daily discharge was measured at the gauges. Missing data could also be estimated from correlations with the other gauges, if there was any correlation.
- P. 5, l. 10ff: Describe in more detail the sediment load data. How was sediment concentration measured by the MWIE? Line 14: I guess you mean 1994 instead of 1964 (also in Fig. 6)?
- P. 5, l. 29: Change "was designed" in "was adapted/changed", since the Watem/Sedem model was also originally developed for temperate climate. Why is the Watem/Sedem model mentioned here? You want to explain why you used the PED model.
- P. 6, l.4: The PED is not a "physically based" model! Change this into "conceptual".
- P. 6, l. 9: Change "Table S1" into "Table 1", delete "with model performance statistics in the supplementary material").
- P.6, l. 21: Explain the three areas in more detail: Which land uses or topographic and soil conditions do they represent? What is meant by "degraded" areas?
- P. 7, l. 4: Give a reference for the Thornthwaite-Mather procedure.
- P. 7, l. 14 ff: Write full sentences to explain the equations!
- P. 7, equation 6: $B_{st}$ instead of $R_{st}$
- P. 8, l 9f: The parameter H needs to be explained in more detail. What is given in Fig. S7? How were the values in S7 determined by field observations? Why is transport capacity limited in the time of highest rainfall?
- P. 8, l. 16ff: The procedure described her is not clear. For calibration you cannot simply take values of another day that fit better to the observed ones!
- P. 8, l 26: You write "initial values were changed systematically…". Please explain in more detail what is meant by systematically here. By which increments were the parameters varied? How many model runs were done?
- P. 8. L. 27ff: Which period was used for calibration of the sediment model? How was "the best fit" determined here?
- P. 9, l. 8ff: This information needs to be given in the Methods section before explaining the calibration procedure.
- P. 9, l. 15/16: "…caused by our inability to estimate amounts of rainfall accurately due to the sparse rain gauge network." This cannot be understood, since the rain gauge network was not described in the Methods section. The authors need to proof that the rainfall data they used is the best available estimate, otherwise modeled discharge can only be not correct! "There was also an anomaly in the collected data in 1996." What do the authors want to express by this? If data are known to be not correct, they should not be used!
- P. 9, l. 19/20: What is meant by "initially" and "later on"? Reformulate this to be more precise.
- P. 9, l. 25: Before you explained that missing precipitation data were filled, so this cannot be an argument for better model performance.
- P. 9, l. 27ff: It needs to be explained why parameters in the sub catchments are different. I would expect them to be different, if the hydrologic conditions (e.g. slope, soil, land use) are different in the sub catchment. L. 29: "Despite that" it the wrong argumentation, since you used the best fitting parameters for each sub catchment.
- P. 10, l. 6: The "unlikely reduction" needs to be explained in more detail.
- P. 10. L. 19: You have to be sure that your measured data are correct, otherwise they cannot be used for model calibration and validation. Data that were not measured "correctly" need to be excluded from further analyses.
- P. 10, l. 22: Delete "in our paper".
- P. 11, l. 9/10: This is not true and needs to be described more differentiated for the different sub catchments, e.g. for Gumara highest runoff is in July whereas highest sediment concentration is in August. Does fig. 6 really show the measured sediment concentrations? From Fig. 4S one can learn that there were only 8 measurements throughout the measurement period (1964.2009 or 1994-2009???). How can monthly means be calculated from these data?

- P 12, l 13ff: The method of calculating the sediment budget of Lake Tana needs to be explained in the Methods section and not in the Results section.
- P. 13, l. 29/30: All available measurements need to be explained in the Methods section.
- P. 14, l 5 and l 26: It needs to be explained or references should be given about what degradation means in the study area.
- P. 15, l. 1-3: This cannot be understood.

*Figures and tables:*

- Fig. 1: In general, the figure looks good! However, please additionally include the gauging stations as a point layer and also the outlet to the Blue Nile. Topography in the background would also be helpful for readers who don't know the area, but might make the figure hard to read. Please color the lake in blue. Eliminate the heading within the map ("Lake Tana Sub Basin"). In the figure caption please mention the method used for river network and watershed delineation (it looks like it was done in SWAT?). Or even describe this in the Methods section because the size of the watersheds is an important parameter in calculating the sediment budget.
- Fig. 2-5: For each of the figures there are no references to figures c and d within the text. Figures c and d should be deleted since the information on the coefficient of correlation is also given in table 3. In this table also the slope and intercept as well as the significance if $R^2$ should be given. However, the regression lines seem to be forced to go through the point of origin which is not appropriate. Please recalculate the regression lines without forcing them through zero. The number of values n needs to be given for each sub catchment. Is it really necessary to show the hydrographs of each sub catchment?
- Fig. 6: How can monthly values be calculated from only a view measurements throughout the measurement period?
- Fig. 7 and figures S4 – S6: The figures can be deleted since the information is also given in table 5. In this table also the slope and intercept as well as the significance if $R^2$ should be given. However, the regression lines seem to be forced to go through the point of origin which is not appropriate. Please recalculate the regression lines without forcing them through zero. The number of values n needs to be given for each sub catchment.
- Fig. S1: Can be deleted, information is given in table 6.
- Fig. S2 and S3: These figures are not very helpful since the differences within the sub catchments are not used for discussion. Instead of land use the proportions of degraded land might be more helpful.
- Fig. S7: This figure can be deleted. The used values can be explained in one sentence within the manuscript.
- Table 1: All abbreviations need to be explained in the heading or footnote. Footnote 3 does not occur within the table. For the study of Easton et al 2010 no validation values are given, hence the study cannot be used for model comparison. The calibration values should always be good; otherwise the calibration would not be good.
- Table 2: Change the heading in "Calibrated PED parameters for calculating discharge and sediment concentration…"

- Table 3: All abbreviations need to be explained in the heading or footnote. RVE and Pbias are not mentioned in the methods section. The number of values n needs to be given for each sub catchment. For Gilgel Abay $R^2$ and NSE of the calibration data set are exactly the same. Is this correct? I guess the monthly values are monthly sums, then the unit is mm/month. This needs to be included in the heading.
- Table 4: Change the heading in "Calibrated sediment concentration parameters …".
- Table 5: All abbreviations need to be explained in the heading or footnote. What is the "error"? It is not mentioned in the methods section. The number of values n needs to be given for each sub catchment.
- Table 6: Change the heading in "Distribution of slope classes in …".
- Table S1: This table could be included directly in the manuscript, if this is the only figure or table left in the supplementary material.

---

## Referee Comment (RC2) · Anonymous Referee #2 · 9 May 2016

The present manuscript on studying sediment budget in the Lake Tana Watershed in Ethiopia was reviewed. The topic is attractive but does not fit to the context well. To me, the manuscript suffers from many deficiencies which discourage me to accept it. There was no comprehensive reviewing of literatures. No justification has been given on the necessity of the work while at least 5 very good paper written by Setegn et al. for the same area. The entire manuscript is very sparse and not connected to each others well. The study is mainly based on models which either need very complicated inputs or very sophisticated to be conceptualized for the study area. The rates given for soil erosion and sediment yield seem to me abnormal and even not corresponded with the data resulted from bathymetric studies. In overall, the present manuscript looks a general report which does not deserve publication in a scientific journal. I hope my comments and suggestions would help authors in their future works. Best Wishes

[Figure]

Please also note the supplement to this comment:
http://www.soil-discuss.net/soil-2015-84/soil-2015-84-RC2-supplement.pdf

**Supplement:**

**Calculating the sediment budget of a tropical lake in the Blue Nile basin: Lake Tana**

F. A. Zimale1, M. A. Moges1, M. L. Alemu1, E. K. Ayana2,3, S. S. Demissie4, S. A. Tilahun2, T. S.

5 Steenhuis2,5\*

[revised manuscript text omitted]

---

## Author Comment (AC1) · 13 May 2016

We are disappointed with the review. Attached is an honest (and not very tactful) response to this review. The reviewer has to option to reply to this harsh response. The discussion period is not closed yet. We are submitting our response three days after the review was posted so the reviewer can respond

Please also note the supplement to this comment:
http://www.soil-discuss.net/soil-2015-84/soil-2015-84-AC1-supplement.pdf

[Figure]

**Supplement:**

We are disappointed with the review. Below is an honest (and not very tactful) response to this review. The reviewer has to option to reply to this harsh response. The discussion period is not closed yet. We are submitting our response three days after the review was posted so the reviewer can respond.

The anonymous review of the rejection of our manuscript is as follows

> "The present manuscript on studying sediment budget in the Lake Tana Watershed in Ethiopia was reviewed. The topic is attractive but does not fit to the context well. To me, the manuscript suffers from many deficiencies which discourage me to accept it. There was no comprehensive reviewing of literatures. No justification has been given on the necessity of the work while at least 5 very good paper written by Setegn et al. for the same area. The entire manuscript is very sparse and not connected to each others well. The study is mainly based on models which either need very complicated inputs or very sophisticated to be conceptualized for the study area. The rates given for soil erosion and sediment yield seem to me abnormal and even not corresponded with the data resulted from bathymetric studies. In overall, the present manuscript looks a general report which does not deserve publication in a scientific journal. I hope my comments and suggestions would help authors in their future works. Best Wishes."

The response to the most blatant inaccuracies in this review follows.

**Comment**
The present manuscript on studying sediment budget in the Lake Tana Watershed in Ethiopia was reviewed.
**Response**
We noted that there were not any posted comments in the annotated manuscript after page 8 and very few (2-3 average per page) on the first eight pages.  We as authors expect a bit more substance why our manuscript was rejected. At least indicate one analysis that is scientifically incorrect)

**Comment**
To me, the manuscript suffers from many deficiencies which discourage me to accept it.
**Response**
The only critical comment on the posted notes on the annotated manuscript was that the sediment measurement for the rating curves were not accurate. We are aware that the sediment rating curves are not accurate. We (Mogus et al) just wrote a paper about that and will likely be published in this journal, but the measurement itself are accurate.

If the reviewer has information to the contrary he/she should state the source. Since there were no comments given after page 8 and very few comments before we are not sure what the "many deficiencies" that are noted by the reviewer to reject the paper.

**Comment**
"No justification has been given on the necessity of the work while at least 5 very good paper written by Setegn et al. for the same area"
**Response**
We cite the paper in the manuscript by Setegn about the Lake Tana basin and that is about hydrology only. There is another paper by Setegn that estimates the sediment load in the Anjeni watershed that has a number of years with detailed hydrology and sediment data. Moreover, most of the papers of Setegn were based on the SWAT model that has not performed well for the Ethiopian highlands. Tilahun et al 2013 (cited in the manuscript) tested the PED model of the same Anjeni watershed and it performed better than the SWAT model with many fewer parameters.

In the manuscript we cite most papers that have been written on the hydrology of Lake Tana and some of the other tropical lakes. This paper present the sediment budget for a tropical lake.

**Comment**
   "The rates given for soil erosion and sediment yield seem to me abnormal and
   even not corresponded with the data resulted from bathymetric studies"
**Response**
This remark makes us even more suspicious how much the reviewer has read of the manuscript since we used the bathymetric data in order to estimate the amount of sediment deposited in the lake. These amounts are in line with what we are predicting. The bathymetric data is described throughout the manuscript, but mostly after page 8 (that did not have posted notes in the annotated manuscript).

For example in the abstract we write (page line 24);

   "Sediment retained in the lake is calculated from two bathymetric taken 15 years
   apart and the sediment leaving the lake is based on measured discharge and
   observed sediment concentrations"

Moreover the soil loss predictions of the model were validated with the data available. Since only a part of the Lake Tana basin was monitored the amount that reaches the lake has a large margin. Finally our rates are in agreement with other studies in the Ethiopian highlands. Where did this reviewer obtain his data to make this statement?

**Comment**
I hope my comments and suggestions would help authors in their future works
**Response**
No these comments were not helpful and there were not any suggestion made. As I indicated we are very much disappointed with this review.

Finally I would like to add that the reviewer writes on two occasions "to me…". This indicates that it is a personal opinion and not based on any literature findings. So we are not sure how to take this rejection of the anonymous reviewer. However, we are sure that most scientific argument likely will not be accepted by this reviewer, because it is not how he/she personally thinks

We are just not very lucky with the reviewers for this paper. We thought that the first reviewer should have been "major revision" but not "a reject". This reviewer suggest also a "reject" based on a ten line review. We frankly have given up. "Soil" must not be the right avenue for this type of paper.

We would like to add that our disappointment with the reviewers should not negatively reflect on the "Soil" journal. We really are very satisfied with the handling and fairness of the comments of the other manuscripts that we have submitted. The "Soil" journal is excellent and we as authors are disappointed that this paper has to be published elsewhere

With high regards

Tammo Steenhuis

---

## Author Comment (AC2) · 13 May 2016

We thank the reviewer for his extensive specific comments. We appreciate the reviewer's suggestion that the manuscript addresses an important topic. We agree that revisions are in order. Many of the points noted in the specific comments will improve the manuscript greatly.

Please also note the supplement to this comment:
http://www.soil-discuss.net/soil-2015-84/soil-2015-84-AC2-supplement.pdf

[Figure]

**Supplement:**

**Response to the Reviewer**
**Soil-2015-84**
**Calculating the Sediment Budget of a Tropical Lake in the Blue Nile Basin: Lake Tana**
**F. A. Zimale, M. A. Mogus, M. L. Alemu, E. K. Ayana, S. S. Demissie, S. A. Tilahun, and T. S. Steenhuis**

We thank the reviewer for his extensive specific comments. We appreciate the reviewer's suggestion that the manuscript addresses an important topic. We agree that revisions are in order. Many of the points noted in the specific comments will improve the manuscript greatly.

However, we find that the reviewer's evaluation that the paper should be rejected is unduly harsh and unfair. Substantive flaws were not found given that purpose was to obtain a sediment balance for Lake Tana with the limited data available in the Lake Tana basin in Ethiopia.

Below we respond to the main reasons for rejection. We clearly show that the comments for rejection of a manuscript are unjustified. In our reaction to the comments, we first repeat the comment of the reviewer and then give our response just below the comment

**Reviewer's Comment 1**
The objectives in the introduction need to be put clearer. A hypothesis was not formulated.

**Response**
A hypothesis is not a requirement for a manuscript as is clearly indicated by a manual on writing and two randomly selected papers cited in "Soil"

The extensive scientific writing manual on the Wiley website by Cargill and O'Connor (2009 p. 50) states the following concerning the end of the introduction (and does not state that a hypothesis is required)

> "At the end of the introduction authors set up the readers' expectations of the rest of the paper they tell them what they can expect to learn about the research being researched"

In addition we looked up the first two papers that were featured on Feb 16 on the "Soil" webpage. Below we cite the text that introduces the rest of their papers. In the manuscript of Groeningen et al. (2015) we find in their manuscript on the nitrogen cycle the following:

"Here, we review important insights with respect to the soil N cycle that have been made over the last decade and present our view on the key challenges of future soil research (Fig. 1). The approach adopted in this paper is three-fold:"

Ping et al. (2015) in a paper on permafrost states:

"In this review, we highlight and discuss important factors affecting the patterns, processes, and carbon stocks of permafrost soils, and summarize recent research developments.

,
The authors of these two papers (chosen at random) and the scientific writing manual are in agreement that a hypothesis the exception rather than the rule.

Our objective as introduction to rest of the paper is clearly understandable

"The objective of this study is to combine current knowledge on sediment transport and to quantify the sediment budget for the Lake Tana and its watershed"

However, reading over the introduction, a valid criticism is that we did not introduce the article sufficiently.

**Reviewer's Comment 2**
It seems that measured precipitation, discharge and sediment data were not analyzed for consistency sufficiently. Measured data that are proven to be not correct need to be excluded or corrected before further analysis and cannot be used for model calibration and validation. The authors should redo the analyzation of the used data. After that the results might be different from the results presented here.

**Response**
The precipitation is highly variable, gages are far apart and even over a 200 m distance rain varies greatly (Bayabil et al. 2016). The discharge changes due to development in the watershed and it is difficult to decide a priori what the cause of any of the slight deviations.

It is very difficult under these circumstances in developing countries to use techniques developed for more developed countries with a temperate climate and less rugged terrain to decide with any certainty the input data that should not be considered (as suggested by the reviewer). With limited data availability, how does the reviewer propose to check the data other than for obvious outliers? In our opinion, suggesting a different approach is appreciated.

It is common practice that this suggestion is accompanied with a specific procedure that has been used in the refereed literature or that the reviewer is familiar with. Without specifics, this comment is superfluous and we cannot comment on it. It certainly should not be used to reject a manuscript.

Finally we would like to point out that most of the hydrological data used by us for obtaining the sediment balance has been used in the refereed literature and apparently of sufficient quality to warrant publication. These publications are: Chebud and and Melesse (2009); Easton et al. (2010; 2012); Gebrehiwot et al (2014); Haile et al (2009); McCartney et al (2010); Rientjes et al (2011); Setegn et al. (2009, 2010, 2011) and Wale et al. (2009)

**Reviewer's Comment 3**
The manuscript needs a major restructuring. Methods and results need to be strictly differentiated.

**Response**
The above comment is the reviewer's perspective. However, there are various ways that the methods and results can be written. As an example, we cite below the style manual by Cargill and O'Connor (2009) concerning the "methods" section'

> "Traditionally, students are taught that the Methods section provides the information needed for another competent scientist to repeat the work. In your experience of reading papers, is this what you find? Many participants in workshops we have conducted report that they have had problems in replicating what authors have done in their published studies even after reading the Methods section thoroughly.

> Another way to think about the goal of the Methods section is that it establishes credibility for the results and should therefore provide enough information about how the work was done for readers to evaluate the results; i.e. to decide for themselves whether the results actually mean what the author claims they mean. Referees are likely to look in this section for evidence to answer the question: Do the methods and the treatment of results conform to acceptable scientific standards?"

This is not an experimental paper where this requirement of differentiation is very important. We give sufficient detail in the material and methods section (and establish credibility for the result section) for the reader to follow the remaining part of the paper. Revisions yes, but rejection is unwarranted in our opinion!

**Reviewer's Comment 4**

The description of some methods and used data is missing (see Specific comments for details) so that some argumentation cannot be understood.

**Response**

We will add these details to the description the methods and used data. Is this really sufficient for a rejection of a paper when **some** methods and used data are missing?

**Reviewer's Comment 5**

The results should be discussed in more detail and should not be related to errors/anomalies in the measured data.

**Response**

This, again, is the perspective of the reviewer. We do not understand why we cannot point out inconsistencies in the data in the results section? This information is helpful to the readers that will use the same data.

Moreover, our findings "in errors/anomalies" in the measured data are significant and worthy to be shown. For example, we are the first among all the watershed models that were run in the Lake Tana basin (see for citations in response to Comment 2) to show that the gage of the Rib overflowed about a certain level and the discharge data were not accurate above that level. This is an important result that should not be hidden from future research.

In addition, the discussion of closing the water balance is extremely important in order to assess if there are other ways that the water can leave the watershed.  It might be of interest that after submitting this manuscript that in one of the recently monitored watershed in the Rib catchment, only 10 % of the rainfall reached the outlet.  Thus, in this case that the water balance is not closed (as often is given as a reason to reject data) would have been sufficient reason to throw out a priori perfectly good data.

Finally, what is the difference if we assess the quality of the data in the result section of the manuscript or beforehand (as suggested by the reviewer)?  A simulation model is basically a mathematical construct that relates the input (rainfall) to the output (stream flow) and is a good way to assess the quality of the data.  Any technique applied a priori essentially follows the same procedure and at the same time assures that the model will fit the data well, so it can be published easily. However, the disadvantage is that potentially information can be lost on functioning of the watershed not foreseen by the model.

**Reviewer's Comment 6**

Figures and tables in the supplementary material are either superfluous or could be integrated directly in the manuscript (see Specific comments for details).

**Response**

The purpose of the supplementary material is to give information to interested readers. We are aware that some might find it "superfluous", but others might find it interesting. We will consider the suggestion of the reviewer, but is this really the right argument to reject the paper for publication?

**In summary**

This paper is about obtaining the best information from a very limited data set in order to provide an answer to an important problem of the sedimentation of Lake Tana. Standard procedures in analyzing the data as proposed by the reviewer are difficult to use in developing countries where data are limited and are collected under political unstable conditions. The other reason given about the form of the manuscript is not shared by all.

This is a research paper where we can redefine procedures to fit better the situation. Using the argument that we do not apply the standard procedures can simply not be a reason to reject a research paper.

There are a number of excellent specific comments given by the reviewer. We will make the improvements, but will not answer in detail below since it is unlikely that the manuscript will be accepted in "SOIL" after we received the 10-line review of referee 2 stating that: "The topic is attractive but does not fit to the context well. To me, the manuscript suffers from many deficiencies which discourage me to accept it.". The deficiencies were not specified.

Finally we are fully aware that as authors, we are in a weak position to make the arguments given in this response to the reviewer's comments and likely will have to withdraw the paper and submit it elsewhere.

Regards

Tammo, Fasikaw, Essayas, Seifu, Dessalegn and Mamaru

**References**

Bayabil HK, Tebebu TY, Stoof CR, Steenhuis TS. Effects of a deep rooted crop and soil amended with charcoal on spatial and temporal runoff patterns in a degrading tropical highland watershed Hydrol. Earth Syst. Sci. in press, 2016

Cargill M. and O'Connor P. Writing Scientific Research Articles" Strategy and Steps. Wiley Blackwell, 2009.

Chebud, YA, and Melesse AM. "Numerical modeling of the groundwater flow system of the Gumara sub-basin in Lake Tana basin, Ethiopia." *Hydrological processes* 23: 3694-3704, 2009

Easton, ZM, Awulachew SB, Steenhuis TS, Habte SA, Zemadim B, Seleshi Y, Bashar KE. Hydrological processes in the Blue Nile. pp 84-111. In S.B. Awulachew, V. Smakhtin, D. Molden and D. Peden, Eds. The Nile River Basin: Water, Agriculture, Governance and Livelihoods. Earthscan from Routledge. New York. 2012

Easton ZM, Fuka DR, White ED, Collick AS, Asharge B, McCartney M, Awulachew SB, Ahmed AA, Steenhuis TS.. A multi basin SWAT model analysis of runoff and sedimentation in the Blue Nile, Ethiopia. Hydrol. Earth Syst. Sci. 14: 1827-1841, 2010

Gebrehiwot, SG, Gärdenäs, AI, Bewket W, Seibert J, Ilstedt U and Bishop K. "The long-term hydrology of East Africa's water tower: statistical change detection in the watersheds of the Abbay Basin." Regional environmental change 14: 321-331, 2014

Groenigen JW, Huygens D, Boeckx P, Kuyper ThW, Lubbers IM, T. Rütting T and Groffman PM. The soil N cycle: new insights and key challenges SOIL 1: 235-256, 2015

Haile AT, Rientjes THM, Gieske ASM and Gebremichael M. Rainfall variability over mountainous and adjacent lake areas: the case of Lake Tana basin at the source of the Blue Nile River." Journal of Applied Meteorology and Climatology 48: 1696-1717, 2009.

Kebede S, Travi Y, Alemayehu T, Marc V. Water balance of Lake Tana and its sensitivity to fluctuations in rainfall, Blue Nile basin, Ethiopia. Journal of hydrology 316: 233-247. 2006.

McCartney, MP, ALemayehu T, Shieraw A and Awulachew SB. Evaluation of current and future water resources development in the Lake Tana Basin, Ethiopia. IWMI Research Report 134. 39 pp, 2010.

Ping CL Jastrow JD, Jorgenson MT, Michaelson GJ and ShurYL. 2015. Permafrost soils and carbon cycling. SOIL 1: 147-171, 2015

Rientjes THM, Perera, BUJ, Haile AT, Reggiani P, and Muthuwatta L P. Regionalisation for lake level simulation – the case of Lake Tana in the Upper Blue Nile, Ethiopia, Hydrol. Earth Syst. Sci. 15: 1167-1183, 2011
.
Setegn SG, Rayner D, Melesse AM, Dargahi B, Srinivasan R. 2011. Impact of climate change on the hydroclimatology of Lake Tana Basin, Ethiopia. Water Resources Research 47: W04511, 2011.

Setegn SG, Srinivasan R, Dargahi B, Melesse AM. Spatial delineation of soil erosion vulnerability in the Lake Tana Basin, Ethiopia. Hydrological Processes 23: 3738-3750, 2009

Setegn SG, Srinivasan R, Melesse AM, Dargahi B. SWAT model application and prediction uncertainty analysis in the Lake Tana Basin, Ethiopia. Hydrological Processes, 24: 357-367, 2010

Wale A, Rientjes THM, Gieske ASM, and Getachew HA. Ungauged catchment contributions to Lake Tana's water balance. Hydrological processes 23: 3682-3693, 2009